# *Lacticaseibacillus rhamnosus* GG Versus Placebo for Eradication of Vancomycin-Resistant *Enterococcus faecium* in Intestinal Carriers: A Systematic Review and Meta-Analysis

**DOI:** 10.3390/microorganisms11112804

**Published:** 2023-11-19

**Authors:** Ingrid Maria Cecilia Rubin, Maja Johanne Søndergaard Knudsen, Sofie Ingdam Halkjær, Christian Schaadt Ilsby, Mette Pinholt, Andreas Munk Petersen

**Affiliations:** 1Department of Clinical Microbiology, Copenhagen University Hospital—Amager and Hvidovre, 2650 Hvidovre, Denmark; mjsk@dadlnet.dk (M.J.S.K.); christian.schaadt.ilsby@regionh.dk (C.S.I.); mette.pinholt@regionh.dk (M.P.); andreas.munk.petersen@regionh.dk (A.M.P.); 2Gastrounit, Medical Section, Copenhagen University Hospital—Amager and Hvidovre, 2650 Hvidovre, Denmark; sofie.ingdam.halkjaer@regionh.dk; 3Department of Clinical Medicine, University of Copenhagen, 1172 Copenhagen, Denmark

**Keywords:** VREfm, LGG, eradication, carriers, systematic review, meta-analysis

## Abstract

The aim of this review was to assess the efficacy and safety of *Lacticaseibacillus rhamnosus* GG (LGG) (previously known as *Lactobacillus rhamnosus* GG) for the eradication of vancomycin-resistant *Enterococcus faecium* (VREfm) in colonized carriers. We searched Cochrane Central, EMBASE, and the PubMed Library from inception to 21 August 2023, for randomized controlled trials (RCTs) investigating the effectiveness of LGG for the eradication of gastrointestinal carriage of VREfm. An initial screening was performed followed by a full-text evaluation of the papers. Out of 4076 articles in the original screening, six RCTs (167 participants) were included in the review. All were placebo-controlled RCTs. The meta-analysis was inconclusive with regard to the effect of LGG for clearing VREfm colonization. The overall quality of the evidence was low due to inconsistency and the small number of patients in the trials. We found insufficient evidence to support the use of LGG for the eradication of VREfm in colonized carriers. There is a need for larger RCTs with a standardized formulation and dosage of LGG in future trials.

## 1. Introduction

Antimicrobial resistance is a growing public health concern, and effective antibiotics are needed to prevent and treat bacterial infections. Antimicrobial resistance may cause treatment failure leading to prolonged illness, mortality, and increased health-care costs [1].

*Enterococcus faecium* has emerged from a commensal of the gastrointestinal tract to the second- or third-leading cause of hospital-acquired infections in the United States and in Europe [2,3]. *Enterococcus faecium* exhibits decreased susceptibility to penicillin and ampicillin, and resistance to most cephalosporins. Additionally, during the last two decades, acquired resistance to vancomycin has been observed [4,5]. This rise in antibiotic resistance seriously limits therapeutic options, when infections occur. Recently, Cassini and colleagues reported 15,917 infections and 1065 attributable deaths caused by vancomycin-resistant *Enterococcus faecium* (VREfm) in Europe in 2015, which is a doubling compared to 2007 [6]. The 30-day mortality of *Enterococcus faecium* bacteremia is high, above 30% for vancomycin-susceptible isolates and 40% for resistant isolates [4,7,8].

Molecular epidemiological analyses revealed the existence of a subpopulation of *Enterococcus faecium* that was associated with hospital-acquired *Enterococcus faecium* isolates. The successful establishment of this *Enterococcus faecium* subpopulation is most likely related to the acquisition of adaptive mechanisms such as antimicrobial resistance genes, virulence determinants, and the ability to survive in the harsh hospital environment [9]. Vancomycin resistance is mediated through different genotypes located on plasmids or in the chromosome of a gene cluster, *vanA* to *vanN* [10]. A variety of different VREfm genotypes have emerged as the bacterium has a high plasticity and acquires plasmids and mobile genetic elements rapidly [11]. Through this genome plasticity and by forming selective adaptive traits, successful clones readily expand within the hospital. VREfm also causes hospital outbreaks, where it spreads between patients, or through the hospital staff and the hospital environment [12].

Previous trials have reported that for each patient with invasive VREfm infection, there are two to ten fecal carriers [13,14,15]. We know that patients colonized with VREfm are at high risk of invasive infections with this bacterium compared to non-colonized patients [16].

In humans, intestinal colonization with hospital-acquired VREfm is thought to be facilitated through disruption of the commensal microbiota of the gastrointestinal tract. A healthy microbiota is assumed to protect against overgrowth with opportunistic pathogens such as VREfm. Previous studies have demonstrated that antibiotic treatment leads to an overgrowth of VREfm [16,17,18].

Antibiotics provoke depletion of the commensal microbiota and are considered the key factor in the disruption of the microbiota [19]. Colonization pressure and environmental contamination are other important factors for VREfm colonization [20,21]. We know that predisposing factors for VREfm invasive infection include VREfm colonization, prior treatment with antibiotics, severe underlying disease, and length of hospital stay [22].

Regarding the natural history of VREfm colonization, results from the literature vary greatly. In a systematic review by Shenoy et al., using logistic regression, 50% of subjects cleared VREfm colonization at 25 weeks after initial colonization [23]. In the study by Roghmann et al., they showed a natural VREfm clearance of only 33% over a three-year study period [24]. An RCT by our group found that almost 60% of patients in the placebo arm cleared VREfm after four weeks, thus constituting the spontaneous clearance [25].

With the rise in the number of VREfm infected/colonized patients and the increasing prevalence of resistance, it is evident that new non-antibiotic options are needed to reduce the number of VREfm cases [4,5]. Probiotics are “live microorganisms which, when administered in adequate amounts confer a health benefit on the host” [26]. *Lacticaseibacillus rhamnosus* (LGG) is a genus of lactic acid-producing Gram-positive bacteria. LGG is one of the world’s most prevalent probiotic strains and has been used in food and dietary supplements since 1990 [27,28].

The LGG strain is usually provided in capsules, sachets, powders, or added to yogurts. Treatment of VREfm-colonized patients with LGG for two to four weeks may reduce the number of VREfm carriers and could be a potential treatment for the eradication of colonization [29,30,31,32]. It has been hypothesized that probiotics in general may help maintain the diversity of the gut microbiota, and that they are important for the restoration of the gut microbiota after antibiotic therapy [26]. Several mechanisms have been proposed to explain the possible specific effects of LGG in clearing VREfm intestinal carriage, as this probiotic has been tested both in humans and in mouse models. Tytgat and colleagues have shown how mucus-binding pili of LGG can prevent the binding of VREfm to the intestinal mucosa [33]. Regarding lactic acid bacteria in general, they use fermentable carbohydrates to produce lactic acid. Lactic acid in turn has been described to acidify the cytosol for most bacteria, eventually leading to cell death [34]. Suggested general mechanisms of how probiotics might work include the improvement in the intestinal barrier function through the production of anti-microbial substances, effects on the epithelium and mucus lining, competitive exclusion, and possibly immune interaction [33,35]. Probiotics are also proposed to have direct bacteriostatic and bactericidal effects against infectious agents [33].

The primary outcome of this systematic review was whether LGG can eradicate VREfm in intestinal carriers. Secondary outcomes were the effects on the health-related change in quality of life, proportion of people colonized with LGG measured through culture or PCR, the proportion of patients with non-serious adverse events and serious adverse events according to International Conference on Harmonization-Good Clinical Practice, and dropouts due to adverse events.

## 2. Materials and Methods

### 2.1. Protocol

The review protocol was registered on PROSPERO Identifier: CRD42023444735 (https://www.crd.york.ac.uk/prospero/ (accessed on 28 July 2023)).

### 2.2. Search Strategy

We searched Cochrane Central, EMBASE, and the PubMed Library from inception to 21 August 2023. *Lacticaseibacillus rhamnosus* was defined as “Lacticaseibacillus rhamnosus” [Mesh]) or “Probiotics” [Mesh]) or “Lacticaseibacillus rhamnosus” [Text word] or Probiotic*[Text word] or “lactobacillus rhamnosus” [Text Word]. Vancomycin-resistant Enterococcus faecium was defined as “Vancomycin Resistance” [Mesh] or “Vancomycin-Resistant Enterococci” [Mesh]) or “Enterococcus faecium” [Mesh] or Vancomycin resistan*[Text Word] or Antibiotic resistan*[Text Word] or “Enterococcus faecium” [Text Word] or “E faecium” [Text Word] or Enterococci [Text Word]. For the full search strategy, please refer to Appendix A.

### 2.3. Data Collection and Analysis

Two authors independently selected the studies in a two-stage process; first, a screening of titles and abstracts was performed; secondly, a final decision of inclusion was performed after a full-text evaluation (MJSK and IMCR). Any disagreements were settled by a third author (AMP).

A data extraction protocol from the computer program Covidence (https://www.covidence.org (accessed on 28 July 2023)) was used and the following information was extracted from each trial: (1) Author, year of publication, trial design, and country of study; (2) VREfm clearance at the end of the trial; (3) treatment description (including formulation, duration, and route of administration); (4) reported non-serious adverse events and serious adverse events; and (5) dropouts.

### 2.4. Inclusion/Exclusion of Studies

We included randomized controlled clinical trials (RCTs). The RCT ought to clearly state the efficacy of LGG as a treatment in one of the study arms, and the studies needed to be placebo-controlled.

During the initial screening of records, we excluded reviews, guidance/recommendations, case reports, non-English articles, retrospective/prospective cohort studies, and records unrelated to study subject. Furthermore, studies with a mixture of probiotics including LGG were excluded. The remaining records were included if they met the inclusion criteria during the full-text eligibility assessment. An overview of the screening and inclusion of records can be found in Figure 1 adapted from the PRISMA statement [36].

PRISMA 2009 Checklist statement: The authors have read the PRISMA 2009 Checklist, and the manuscript was prepared and revised according to the PRISMA 2009 Checklist.

### 2.5. Quality Assessment

We adapted the Quality Assessment Tool of the National Institutes of Health for RCTs to assess the risk of bias of the individual studies with 12 different criteria. Criteria 9, 13, and 14 were adapted to include compliancy, no subgroup analysis, and intention-to-treat analysis. Additional criteria 15 and 16 were included to assess specification of dosage and duration of treatment (https://www.nhlbi.nih.gov/health-topics/study-quality-assessment-tools (accessed on 28 July 2023)). For each study and criterion, the bias was reported as high risk, low risk, or unclear risk based on the assessment by two authors (IMCR and MJSK). Again, any disagreement was settled by a third author (AMP). The table used in the bias assessment process is found in Appendix A.

### 2.6. Statistical Analyses

Meta-analysis was performed using R (v. 4.2.2) [37] to assess the risk difference (RD) associated with VRE probiotic treatment across the six included studies. We employed the Mantel–Haenszel method with both fixed- and random-effects models. Given the significant heterogeneity observed (I^2^ = 89.8%, *p* < 0.001), the results from the random-effects model were prioritized. The model was evaluated by performing a funnel plot.

## 3. Results

### 3.1. Study Design and Selection

From the literature search, we identified a total of 5368 articles, which were imported for screening into the computer program Covidence. After removal of duplicates, we identified 4076 articles for the initial screening. Out of these articles, 23 were retrieved for eligibility screening. Of these, 17 were excluded based on the following: five were conference abstracts, two were clinical trial protocols, two were not placebo-controlled trials, one used another probiotic strain, five were not RCTs, and two used a mixture of probiotics. Therefore, a total of six articles, published between 2007 and 2022, met the inclusion criteria and were included in this review. The trials were conducted in Australia [29], the United States [31,38], Denmark [25], Poland [30], and France [39]. An overview of the screening and inclusion of records can be found in Figure 1 adapted from the PRISMA statement [36].

### 3.2. Description of the Studies

All included studies were two-armed RCTs. Five out of the six studies evaluated the effect of LGG on the eradication of VREfm as the primary outcome. One study evaluated the effect of LGG on the prevention of the acquisition of antibiotic-resistant organisms (AROs) (of which VREfm was one) with the eradication of AROs as a secondary outcome. A total number of 167 patients were included in the six studies. Regarding study demographics, five of the studies included an older patient population, with the mean age ranging from 68 to 77 years. Only the study by Szachta et al. included a population of children [30]. Most studies reported a predominance of men [29,30,31,38]. For a full description of the studies, refer to Table 1.

In four studies [25,31,38,39], LGG and placebo were administered as orally taken capsules or capsules dissolved and given via a nasogastric tube [38]. In the remaining two studies, LGG and placebo were provided to the patients as a yogurt in one study [29], as capsules in four studies [25,30,31,38,39], or as a sachet dissolved in water or milk in another study [30]. Five studies specified the daily dose of LGG ranging from 1 billion CFUs to 60 billion CFUs [25,30,31,38,39]. The duration of intervention ranged from 5.8 days to 5 weeks.

Four studies used culture-based methods with susceptibility testing of stool samples or rectal swabs to determine VREfm intestinal carriage. Doron et al. used a decline in colony count to evaluate the clearance of VREfm from the stool [31]. Rubin et al. used a specific PCR for vanA followed by a confirmatory culture [25], while Vidal et al. did not specify their methods for the determination of VREfm carriage [39].

### 3.3. Primary Outcome

#### Meta-Analysis and Efficacy of LGG on Eradication of VREfm from the GI-Tract

The result of the meta-analysis is presented in Figure 2, and the result of the funnel plot is shown in Appendix A. The funnel plot analysis revealed potential bias, as some studies were situated outside the funnel. This confirmed that the meta-analysis could not be used as evidence to assess the effect of LGG on the eradication of VREfm. This could be due to publication bias, or the fact that the studies were significant in opposing results. Thus, the meta-analysis was inconclusive for LGG on the eradication of VREfm in intestinal carriers. We found a risk difference of 0.20 with a wide 95% CI of −0.17–0.57.

Out of the six studies, only the study by Manley et al. presented persisting significant results with regard to LGG on VREfm clearance. In their treatment group, 11/11 patients cleared VREfm after eight weeks, compared to only 1/12 in the placebo group (*p* value not specified in this study) [29].

Szachta et al. reported a temporary effect on VREfm clearance in a population of hospitalized children. At the end of the intervention (three weeks), there was a significantly higher number of children with VREfm clearance in the treatment group (62.5%) than in the placebo group (24%) (*p* = 0.002). However, this difference did not persist at the seven-week follow-up [30].

The other four studies reported no effect of LGG on clearing VREfm.

### 3.4. Secondary Outcomes

Statistical analyses for the secondary outcomes were not deemed possible, due to the lack of data and lack of heterogeneity.

#### 3.4.1. Health-Related Change in the Quality of Life

The health-related change in the quality of life was measured by none of the studies.

#### 3.4.2. LGG Colonization at the End of Intervention

The studies by Doron et al. [31], Szachta et al. [30], and Rubin et al. [25] looked at the LGG or *Lactobacillus* spp. content in stool either by using culture or PCR. All these studies detected LGG or *Lactobacillus* spp. in the stool samples, suggesting they used sufficient doses of LGG. Please refer to Table 1 for a summary of the results.

#### 3.4.3. Tolerability and Safety

Half of the studies [29,30,39] did not report adverse events, and the other half reported neither adverse nor serious adverse events related to the treatment [25,31,38].

#### 3.4.4. Heading

Regarding all-cause mortality, only two studies reported on this. In the study by our group (Rubin et al.), five patients died during the intervention. None of these deaths were related to the LGG treatment [25]. In the study by Manley et al., one patient in the treatment arm died during the intervention [29]. This is not further described in this article. In general, the intervention with the probiotic strain, LGG, was considered safe and well tolerated.

### 3.5. Risk of Bias

The result of the bias assessment is presented in Figure 3. Overall, the studies represented a low risk of bias on most criteria that could be assessed. Criteria with a high risk of bias in all studies were power and sample size.

## 4. Discussion

In this meta-analysis, we could not determine the effect of LGG on the clearing of VREfm from the gastrointestinal tract. The included studies in our systematic review have shown significant and opposing results, or the inconclusiveness could be due to publication bias. Another limitation to our systematic review is that we could only include six studies, and four of these were underpowered. Also, heterogeneity was high as the formulations, dose, and duration of the LGG strain differed, as well as the follow-up time.

Manley et al. presented the only truly positive study [29]. A limitation to their study was that the dose of LGG in the yogurt was not specified. Interestingly, their study was the only study using a yogurt formulation. As has been speculated by others, the formulation of the probiotic strain might indeed impact its effect [31]. However, four of our included studies measured LGG or *Lactobacillus* spp. count from the stool after the intervention, thereby showing that the probiotic did indeed reach the colon. In the population of VREfm-colonized children, Szachta et al. concluded that VREfm clearance was perhaps just suppressed temporarily, as it was not sustained after the intervention was withdrawn [30]. A limitation to their study was a lack of adhesion to the protocol, with a high loss to follow-up of 37%. None of the patients in the study by Doron et al. cleared the VREfm [31]. As the authors point out, their intervention had a duration of only two weeks, and the study by Szachta et al. [30] only saw an effect of LGG by week three. This suggests that the duration of intervention could play a role in clearance. In the study by our group (Rubin et al. [25]), we showed no significance between the placebo group and the control group, and in both groups, almost 60% of the patients had cleared VREfm after the 4-week intervention. At the 24-week follow-up, almost 90% had cleared VREfm in both groups, thus implying a higher natural decolonization than previously shown. Szachta et al. [30] showed a natural clearance of VREfm of 33% in the placebo group, which is in line with previous studies [24]. This leads us to suspect that discharge from the hospital itself could play a role in the decolonization of VREfm, as it will revert the dysbiosis of the gut microbiome (discussed further down).

Interestingly, during the screening process, we found a study by Buyukeren et al., showing a positive effect of 1 × 10^9^ CFU/g of LGG in VREfm-colonized neonates. Their study showed a significant difference in decolonization between the arms, as 21/22 neonates cleared VREfm in the LGG arm vs. 12/23 in the control arm (*p* < 0.05) [40]. They also performed a six-month follow-up, where 11 patients in the placebo arm remained VREfm-colonized vs. only one in the LGG arm. However, their study was not placebo-controlled, and it could be biased in that parents with more critically ill neonates opted out of the study, making the arms non-comparable. Furthermore, the duration of treatment varied as it was withdrawn after three consecutive negative results. The planned treatment was six months. Also, it is difficult to compare a study of neonates with adults or even children, as their microbiota is vastly different and dramatically changes during the first year of life [41].

A mini-review by Crouzet et al. investigated various probiotic strains or mixtures thereof on their anti-VREfm effect [42]. They concluded that some strains, including LGG, might reduce the intensity of VREfm colonization, but some bacterial reservoirs might remain, and their conclusion was that the eradication of VREfm with only probiotics might be difficult to achieve.

In future studies with probiotics for the eradication of VREfm, a prospective cohort study design might be favorable over the RCTs. Carriers of VREfm in general, and most participants in these studies, appear to be recruited from a geriatric population and, thus, presumably a comorbid population. Both the study by Doron et al. and the study by our group (Rubin et al. [25]) had difficulties in recruiting the required number of participants. In the latter, the most common criterion for exclusion was the inability to sign informed consent, e.g., suffering from dementia. We believe that a prospective cohort study design such as the study by Rauseo et al. [38] would overcome these recruitment challenges. The primary outcome of a prospective cohort study design would be the incidence of VREfm acquisition during hospitalization. Here, a set-up with a comparison of different wards with equal VREfm burden could be investigated.

Multiple studies have investigated the effect of probiotics on preventing the acquisition of multidrug-resistant organisms in hospitalized patients. In a prospective cohort study by de Regt and colleagues, they screened all patients in their ward for ampicillin-resistant *Enterococcus faecium* (AREfm) twice weekly [43]. They administered a mix of probiotics to all included patients twice daily. The mix consisted of *Bifidobacterium bifidum*, *Bifidobacterium lactis*, *Enterococcus faecium*, *Lactobacillus acidophilus*, *Lactobacillus paracasei*, *Lactobacillus plantarum*, *Lacticaseibacillus rhamnosus*, and *Lactobacillus salivarius* with a total concentration of 10^9^ CFU/g. At the end of the study a with cross-over design, the authors had not observed a reduction in acquisition of AREfm. In a different study by Borgmann and colleagues, they administered two probiotics to all patients receiving antibiotics in an early rehabilitation ward [44]. In this ward, patients with stroke and trauma injuries were hospitalized. The two probiotics were *Saccharomyces boulardii* in a dose of 375 mg daily and *Escherichia coli* Nissle in a dose of 2.5–25 billion bacteria per capsule twice daily. The intervention followed the closure of the ward due to an outbreak of carbapenem-resistant *Klebsiella pneumoniae*. Patients were screened for methicillin-resistant *Staphylococcus aureus* (MRSA), extended-spectrum beta-lactamase (ESBL)-producing Gram-negative bacteria, and VREfm. The authors observed a decrease in acquisition of VREfm- and ESBL-producing Gram-negative bacteria other than *Escherichia coli* during the intervention period. The prevalence of MRSA- and ESBL-producing *Escherichia coli* did not decrease. In another four-armed placebo-controlled study by Toh and colleagues, they investigated the effect of different probiotics on the prevention or clearance of VREfm [45]. The prevention and clearance of VREfm were secondary outcomes in this study, as the primary outcome was the time interval from randomization to the first symptomatic urinary tract infection in a population of patients with spinal cord injury. They found no effect of probiotics on the prevention or clearance of VREfm.

A further point to be made is that a single bacteria probiotic might not have a sufficient effect to revert the dysbiosis of the gut microbiota. In the study by our group, we did not detect any difference in the microbial diversity in terms of alpha-diversity between the treatment and the placebo arm [25]. The negative effects on the gut microbiota by hospitalization and antibiotics have been observed in many previous studies [46,47,48,49]. In a study by Chanderraj et al., they demonstrated that only 21% of the microbiota remained unchanged at 24 h after hospitalization vs. at admission [46]. Thus, any attempt at reversing the gut dysbiosis through treatment with a single probiotic strain, or even a mixture of strains, might prove inadequate considering the overwhelming changes taking place during hospitalization. The negative effect of antibiotics on the microbiota was highlighted in a study by Donskey et al. where they demonstrated that patients previously colonized with VREfm and who had had three negative rounds of VREfm were, in most cases, re-colonized with VREfm after a new antibiotic regimen [50].

New treatment options for the eradication of VREfm need to be explored in future studies. A study by Dinh and colleagues investigated the effect of fecal microbiota transplantation (FMT) on the eradication of VREfm in intestinal carriers [51]. At the eight-week follow-up, 6/9 patients had eradicated VREfm after FMT. This study had no control group, and the hypothesis needs to be further evaluated in future RCTs.

## 5. Conclusions

We did not find evidence to support the use of LGG for the eradication of VREfm gastrointestinal carriage, although LGG proved to be safe and well tolerated. The meta-analysis was inconclusive. There is a need for a standardized formulation and dosage of LGG in future trials. New treatment options including FMT for the eradication of VREfm in intestinal carriers should be evaluated in future studies.

## Figures and Tables

**Figure 1 microorganisms-11-02804-f001:**
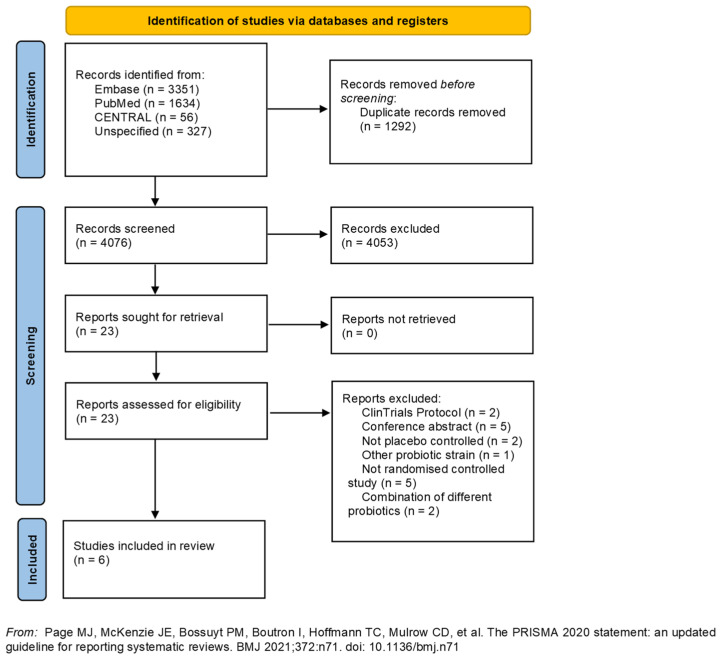
Flowchart of in- and excluded papers [36].

**Figure 2 microorganisms-11-02804-f002:**
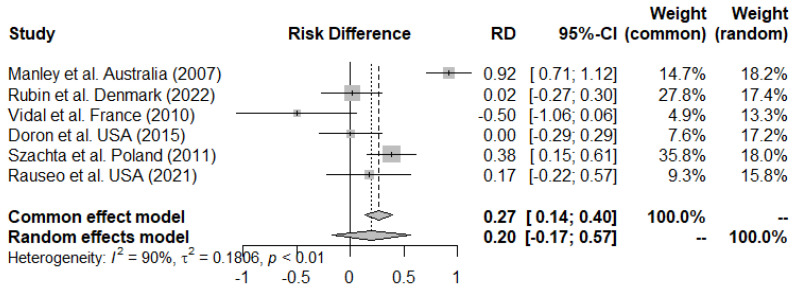
The forest plot presents a meta-analysis of six studies evaluating the risk difference (RD) using the Mantel–Haenszel weighting method. The analysis reveals a high degree of heterogeneity among the studies (I^2^ = 89.8%, *p* < 0.001). A total of 167 observations were included, with 74 events recorded [25,29,30,31,38,39].

**Figure 3 microorganisms-11-02804-f003:**
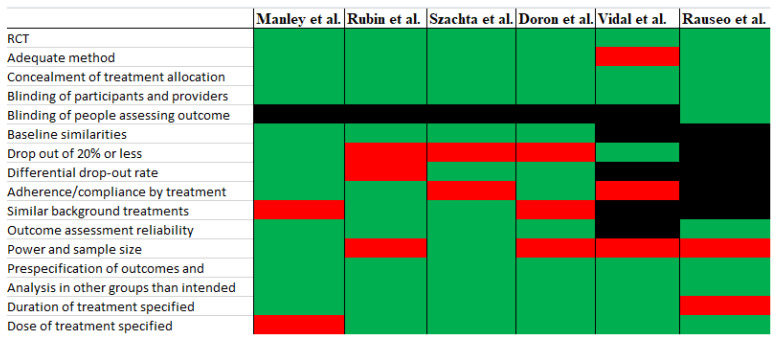
Risk of bias assessment. Green indicates low risk of bias, red indicates high risk of bias, and black indicates unclear risk of bias or could not be assessed [25,29,30,31,38,39].

**Table 1 microorganisms-11-02804-t001:** Summary results of all included papers.

Paper	Study Design	Intervention and Duration	Number of Individuals (LGG:Placebo)	Male:Female Ratio	Mean Age in Years (Range)	Follow-Up	Dose and Formulation of LGG	Number of Cleared Individuals at the End of Intervention	Number of Cleared Individuals at the End of Follow-Up	Adverse Events	Conclusion
Manley et al. Australia (2007) [29]	Double-blinded, randomized and placebo-controlled trial with two arms.	1. LGG yoghurt 2. Placebo; 4 weeks	23 (11:12)	10:4	68 (46–84)	8 weeks	100 g of yoghurt, unknown LGG dose	11/11 in treatment group and 1/12 in placebo group	8/11 in treatment group and 0/8 in placebo group	Not reported	Significant effect of VRE eradication
Vidal et al. France (2010) [39]	Double-blinded, randomized and placebo-controlled pilot trial with two arms.	1. LGG 2. Placebo; 5 weeks	8 (6:2)	not reported	77 (66–92)	11 weeks or until negative VRE culture	1 × 10^9^ CFU of LGG daily as capsules	3/6 in treatment arm and 2/2 in placebo arm	2/4 in treatment group	Not reported	No effect on VRE clearance
Szachta et al. Poland (2011) [30]	Double-blinded, randomized and placebo-controlled trial with two arms.	1. LGG 2. Placebo; 3 weeks	61 (32:29)	40:21	2.5 (unknown)	4 weeks	3 × 10^9^ CFU of LGG daily as a sachet dissolved in water or milk	20/32 in treatment arm and 7/29 in placebo group	10/19 in treatment arm and 9/20 in placebo group	Not reported	Temporary effect of VRE eradication
Doron et al. USA (2015) [31]	Double-blinded, randomized and placebo-controlled trial with two arms.	1. LGG 2. Placebo; 2 weeks	11 (5:6)	7:4	70 (53–90)	2 weeks	2 × 10^10^ CFU of LGG as capsules daily	0/5 in treatment arm and 0/6 in placebo arm	0/4 in treatment arm and 0/5 in placebo arm	No adverse events related to LGG were seen	No effect on VRE clearance
Rauseo et al. USA (2021) [38]	Double-blinded, randomized, and controlled pilot trial with two arms. (VRE clearance = secondary outcome)	1. LGG 2. Placebo; median duration of intervention was 5.8 and 6.5 days for LGG and placebo respectively	16 (7:9)	Not reported for the subgroup of VREfm carriers	Not reported for the subgroup of VREfm carriers	Every 3 days after enrollment and at discharge	2 × 10^10^ CFU of LGG as capsules or nasogastric administration	2/7 in treatment arm and 1/9 in placebo arm	Same as end of intervention	No safety concerns and no difference in Bristol stool types between arms	No effect on VRE clearance
Rubin et al. Denmark (2022) [25]	Double-blinded, randomized and placebo-controlled trial with two arms.	1. LGG 2. Placebo; 4 weeks	48 (21:27)	18:30	75 (64.5–82.5)	24 weeks	6 × 10^10^ CFU of LGG as capsules daily	12/21 in treatment group and 15/27 in placebo group	7/8 in treatment group and 9/10 in placebo group	No adverse events related to LGG were seen	No effect on VRE clearance

## Data Availability

All available data used in this review are available in the original articles.

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
