# Peer review of "Lacticaseibacillus rhamnosus GG Versus Placebo for Eradication of Vancomycin-Resistant Enterococcus faecium in Intestinal Carriers: A Systematic Review and Meta-Analysis"

_microorganisms, 2023, doi:10.3390/microorganisms11112804_

Round 1
Reviewer 1 Report
Comments and Suggestions for Authors
Comments on the Quality of English LanguageMinor editing of English language required
Author Response
Thank you for your thorough review or our manuscript, which we really appreciate. We have responded to each of the points and highlighted the response in the revised manuscript

Reviewer 2 Report
Comments and Suggestions for Authors
Interesting short paper with clinical inclination reviewing efficacy of LGG in eradication of vancomycin-resistant enterococci in randomized clinical trials. The review was performed with good standard methods. Unfortunately, number of the trials was very limited and thus conclusions could not be drawn, but it the eradication seemed to be ineffective.
L.rhamnosus strain (LGG) should be mentioned in the title since this review is devoted to only this strain.
Discussion is unproportionally long and should be tailored.
Gram staining should be always capitalized since it is a name(!) of the Danish microbiologists(!)
Detailed:
-line 45: term “success” sounds improperly in this context
-line 79: LGG is abbreviation of Lactobacillus rhamnosus Goldin & Gorbach who characterized and patented this particular strain but not of other strains in this species. Moreover, the genus Lactobacillus has been divided into several new species. Thus, the authors should use valid name of the species: Lacticaseibacillus rhamnosus, Lacticaseibacillus rhamnosus GG for the strain or LGG in abbreviation along the whole MS.
- 83: The LGG bacteria are …...provided in capsules…
- line 292: fecal microbiota but not microbiome.
Round 2
Reviewer 1 Report
Comments and Suggestions for Authors
The authors responded to all reviewer comment. Except for not correcting the style of citing references within the text. For example Line 36: [2], [3] should be [2, 3].
Comments on the Quality of English LanguageMinor editing of English language required
Author Response
Dear Reviewer,
Thank you for you feed-back. The references should be correct now. Please see the updated manuscript.